# Assessing Access to WASH in Urban Schools during COVID-19 in Kazakhstan: Case Study of Central Kazakhstan

**DOI:** 10.3390/ijerph19116438

**Published:** 2022-05-25

**Authors:** Berik Toleubekov, Zhanerke Bolatova, Martin Stafström

**Affiliations:** 1Division of Social Medicine and Global Health, Department of Clinical Sciences, Lund University, Jan Waldenströms gata 35, 214 28 Malmö, Sweden; martin.stafstrom@med.lu.se; 2School of Public Health, Biomedicine and Pharmacy, Karaganda Medical University, Karaganda 100008, Kazakhstan; bolatovazhanerke93@gmail.com

**Keywords:** WASH, schools, access to WASH, SDG, drinking water, sanitation, hygiene, system approach

## Abstract

The WHO/UNICEF Joint Monitoring Program (JMP) for Water Supply, Sanitation and Hygiene (WASH) produces global estimates of the real situation of access to water, sanitation and hygiene services, and sanitation and hygiene in households, educational institutes and health care facilities; however it is lacking data on schools in Kazakhstan. Thus, the aim of this research was to assess access to WASH in schools of urban area in Kazakhstan. The study was conducted in seven schools of Central Kazakhstan during the COVID-19 pandemic and restrictive measures. Three data collection methods were used: a questionnaire for administrative staff, a questionnaire for parents and observation. Parents of offline study pupils (only second and third grades due to the pandemic) were included in the survey. Students had access to in-building toilets in all schools connected to the centralized sewer. The number of school toilets varied from 7 (KAZ200085) to 61 (KAZ200089). The average amount of toilets was 28.08 ± 16.97. Only two out of seven schools complied with the requirements of Kazakhstan national sanitary standards for the ratio of school toilets to the number of students. From the questionnaire with the school administrations, it was defined that the primary source of drinking water was the public water supply. All schools regularly disinfect and check the water supply system. At the same time, the results also revealed discrepancies in the answers between administration and parents (2.6% of parents showed that their children have rare access to drinking water), and insufficient monitoring of implementation of WASH services. This study also confirmed that the full provision of access to water and water services in the structure of educational institutions solves several SDG targets.

## 1. Introduction

Water, sanitation and hygiene are fundamental human requirements that have an impact on ensuring basic rights to a decent standard of living and health [1]. The lack of data on schools in Kazakhstan in the WHO/UNICEF Joint Monitoring Program (JMP) for WASH (Water, Sanitation and Hygiene) and COVID-19 pandemic emphasizes these fundamental societal needs, especially for millions of school-going children, who potentially may not have access to drinking water and basic hygienic facilities [2,3,4]. Safe drinking water, as one of the WASH criteria, can also provide an assessment of the well-being of individuals and social groups [5]. Assessment of access to drinking water is a measurement that determines further policy and management related to water supply in regions and institutions. This can help governments and oversight agencies budget more accurately for WASH activities in schools and other institutions [6]. Schools, as the primary and one of the most important links in human socialization, require the highest quality and constant monitoring of water services. Improved WASH could save the lives of over 2 million children under the age of 5 around the world every year [7].

Moreover, thorough monitoring and assessment of sanitation and hygiene conditions during the COVID-19 pandemic and lockdown in the Republic of Kazakhstan has proved and raised the level of their significance and magnitude. Due to the high contagiousness and rapid spread of the virus, including through contact with a surface contaminated with SARS-CoV-2 [8,9,10], the pandemic highlighted the need to motivate people to maintain adequate hand hygiene, which includes providing access to appropriate hand washing facilities, including soap [11]. Adequate water, sanitation and hygiene services reduce the potential risks of spreading of COVID-19 or other infections in the school environment. Numerous studies have shown the effectiveness of proper hygiene interventions and appropriate WASH facilities, and also their positive impact on reducing of spreading of infectious diseases and/or absenteeism among students [6,12,13,14,15,16,17,18].

The Sustainable Development Goal 6 highlights the importance of achieving universal and equitable access to safe drinking water and sanitation for all by 2030 [19]. According to SDG 4, everyone should have access to equitable quality education and opportunities for equal and lifelong learning [20]. Student success at school is highly affected by the school environment, which includes health skills and hygiene behavior [21]. All SDG goals interact with each other and jointly set the right vector of development for countries, that are sometimes at different stages. Thus, considering in detail the SDG 4 and SDG 6, their targets and their relationship, one of the tasks of WASH is clearly outlined: the creation of the necessary conditions and equal opportunities for children to receive a quality education. Therefore, access to water, sanitation and hygiene in educational institutions is an important link to achieve the targets of SDG 4 [22], paying special attention to the needs of girls and people living with disabilities, as expressed in the SDGs [19,23].

Accounting for the importance of water services, this paper aims to assess the access to WASH in urban schools of Central Kazakhstan. The study is not only a simple attempt to identify the problems associated with access to WASH, but also to determine where the general understanding of the importance of access to WASH in schools and its direct impact on inclusive and equitable education and the creation of a healthy and prosperous environment is. The collected data and analysis in the future should play a role in the correct setting of targets for achieving the SDGs and for potential research directions [22], in assessing the real situation with water, sanitation and hygiene in schools as objectively as possible.

## 2. Materials and Methods

### 2.1. Study Design

Three data collection methods were used in the survey: a questionnaire for administrative staff, a questionnaire for parents, and observation. The basis for all these tools was the WHO/UNICEF “Surveillance of water, sanitation and hygiene in schools” [23]. The use of the selected tools covers all aspects of WASH and allows us to address the research aim. The purpose of cross-checking the data is to capture different facets of WASH services and different points of view on them, allowing a more realistic look at possible problems.

The questionnaires were validated in the research “Challenges of Access to WASH in Schools in Low- and Middle-Income Countries: Case Study from Rural Central Kazakhstan” [21]. The purpose of a questionnaire for school administrative staff is to obtain data on the management of the school, their expectations regarding the provision of WASH services, and their general understanding of the importance of this area [5]. The survey aims to identify the level of their understanding of the relationship between the quality of water, sanitation, hygiene services and the creation of an environment for ensuring inclusive and quality education [20,22]. An assessment of the administration’s point of view should answer the question of “How seriously do they take WASH-related issues in educational institutions?” [19].

A questionnaire for parents is required to show their views on the implementation of water, sanitation and hygiene programs in schools and level of satisfaction with the services available to their children [5]. An analysis of parent responses should expand the perspective on understanding whether WASH facilities are acceptable and what other challenges lie ahead to create a more comfortable environment for students.

Observation is used to obtain objective information about the real situation on the ground through selective inspections. The purpose of this method is to complete the picture of water, sanitation and hygiene in schools with unbiased data. Each element of water services has an evaluation criterion. According to these assessment criteria, Table 1 shows which tool and which questions were used to assess which WASH criterion.

### 2.2. Study Area

The study was conducted in Karaganda city (Figure 1), located in the Karaganda region/agglomeration (Central Kazakhstan). The climate is humid continental climate. According to the Agency for Strategic planning and reforms of the Republic of Kazakhstan and Bureau of National statistics (https://stat.gov.kz/region/256619), the population of Karaganda was 501,095 accessed on 21 December 2020. The population in the Karaganda region was 1,376,882 in 2020. There are 537 schools on the territory of the Karaganda region, of which 237 are in urban areas, and 66 in the city of Karaganda.

### 2.3. Schools

The schools were divided according to the number of pupils: small, medium and large schools. If the number of students did not exceed 800, then it was a small school, medium schools ranged between 800 to 1200 pupils, and any number above identified a large school. Seven schools were included in the study (see Section 2.5). Schools were chosen randomly. Of the selected schools, three schools were small (KAZ200086—736 students, KAZ200090—609 students, KAZ200085—776 students), two schools were medium (KAZ200084—1038 students, KAZ200089—1023 students), and two out of seven schools were large (KAZ200087—1391 students, KAZ200088—2671 students). Each school taught students in two shifts and the medium of instruction was both Kazakh and Russian.

### 2.4. Data Collection

Initially, a meeting was arranged with the director and/or administration of the school, during which the aim, objectives and importance of the study was explained. Then the school administration filled out a questionnaire for administrative staff, which consisted of 15 questions with various sub-questions and options.

Further, the school administration provided class schedules. With the help of the administration, while consulting the schedules, the research team was able to arrange research data gathering meetings with the parents of students. A total of 450 parents agreed to take part in the meetings, where they were informed about the study by the researchers and questionnaires were distributed. Additionally, the researchers answered the parents’ questions. Before the survey, parents read and signed the consent form about the purpose and importance of the study. Parents who declined to participate returned blank questionnaires. A total of 348 parents (Table 2) completed the survey.

The researchers collected all the questionnaires in the space of one week.

All schools granted permission to the researchers to carry out on site observations, but with some restrictions due to COVID-19 regulations. The observers were only permitted to inspect one toilet at each school. A school administrator accompanied the researcher to each toilet, where the researcher filled out an observation sheet consisting of 16 questions.

### 2.5. Limitations

The data were collected from mid-October to November 2020 during the COVID-19 pandemic and subsequent restrictive measures in Kazakhstan. The study underwent several adjustments in the process and was repeatedly postponed and delayed due to difficulties and national decrees in the wake of the pandemic [24].

First of all, a survey among the students was excluded from the methodology, since students in most grades were studying from home. At the time, the schools had restricted access and only students in grades 1–4 were having classes regularly on site.

Ten schools declined to participate, and of those who agreed to take part only permitted observations to be conducted in a single toilet respectively. Due to these study limitations, it became necessary to only select specific questions from the WHO/UNICEF tool “Surveillance of WASH in schools”, rather than the full protocol. The tool was modified, which means that for some of the questions, answer alternatives and vocabulary were altered to match the setting. The questionnaire was, however, piloted prior to full implementation.

## 3. Results

### 3.1. Water

The functioning and maintenance of water services were controlled by school directors in two schools and head teachers in five. Data from the school administration questionnaire showed that the main source of drinking water supply in all schools was a centralized water supply system connected to the building. All schools carried out regular disinfection and cleaning of water faucets, water purification, and regular checks of the water supply system to identify violations in the provision of drinking water supply. Six out of seven administrators answered that the main source of drinking water was always available throughout the school year. One school did not answer the question. Six out of seven respondents assured that students could always drink water when necessary, including during classes. One respondent did not answer this question.

A total of 49.1% were parents of boys (171 people), and 50.9% of girls (177 people). About 50% (53.2%) of parents answered that their child always had access to drinking water, and mainly (free of charge) during the entire stay at school (Table 3). A very small (2.6%) percentage of parents chose the option “rarely”, whereas 14.1% of parents answered negatively to this question. Furthermore, 7.7% of respondents added comments that they brought water to school, and one parent indicated that the cafeteria was closed. The questionnaires answered by parents were collected during the COVID-19 pandemic with temporary restrictions recommending to close canteens.

During the observation phase, information and education material about water in the schools were seen. This conveyed that pupils could get drinking water from taps or fountains outside the toilet and from the canteen for free. Additionally, students could bring water from home or buy water in the canteen or buffet.

### 3.2. Sanitation

Students had access to a toilet inside the building in all schools, connected to a centralized water supply. Moreover, one of the schools had an additional type of toilet not connected to a centralized sewer (KAZ200084). The number of school toilets varied from 7 (KAZ200085) to 61 (KAZ200089). The average amount of toilets was 28.08 ± 16.97. The proportion of toilets to the number of pupils varied. Kazakhstani national sanitary regulations require at least one toilet for every 20–30 pupils. Only two of the seven schools met this requirement (KAZ 200086 and KAZ 200089), where the proportion was equal to 18.4 (40 toilets to 736 pupils) and 16.8 (61 toilets to 1023 pupils), respectively. The greatest lack of toilet facilities per student was found in a small school, where the proportion was 110.8 (KAZ 200085, 7 school toilets to 776 pupils). The largest school in this study had a student/toilet ratio of 83.5 (KAZ 200088, 32 school toilets). In other schools, the ratio ranged from 43.5 to 57.7.

All school toilets were divided by sex. The maximum number of toilets for girls was 32 (KAZ200089), and 21 for boys. The minimum number of toilets for girls and boys was 3 (KAZ200085). Moreover, almost all schools had toilets for staff, except school KAZ200086. Furthermore, only two schools had urinals; in one of them (KAZ200089), the number of urinals was 16. All respondents answered that school toilets provided sufficient privacy for students, except school KAZ200085. Almost all administrations described that the pupils could use the toilets during the school day whenever needed. An exception was school KAZ200088, where students used the toilets at any time, but during classes they were only available upon request.

According to the school administration questionnaire, two schools had had problems with the functionality of the toilets that had recently been resolved. The five other schools did not have any current problems. Adequate lighting in the toilets was available in all schools according to the same questionnaire. Almost all (5/7) school administrations assured that there was sufficient ventilation, one respondent did not answer, and one school reported the ventilation to be inadequate (KAZ200090). Heating was, according to the data, available in all school toilets; KAZ200086 did not respond to this question. Furthermore, 4/7 schools claimed that toilet paper was always available. It was provided most of the time in one school. One school reported that they did not provide toilet paper. One school did not respond. All school toilets were cleaned twice a day or more often if needed.

About 60% (60.6%) of parents reported that their children could go to the toilet at school whenever they needed, whereas 15.8% of children went to the toilet rarely, only when it was hard to endure. However, 3.4% of parents reported that their children did not use the school toilet facilities. A small (2.9%) percentage of parents did not know if their children used the school toilets or not. About 78% (77.9%) of parents reported that the school toilets were located inside the school. However, 1.7% indicated other options and wrote open ended answers indicating that their children only used the toilet at home. Furthermore, 19.5% of respondents did not respond, and 0.9% of parents did not know what type of toilet their children used.

The observations conducted by the researchers showed that flushing toilets were available. All toilet cabins were accessible, though the students were unable to lock the doors. In school KAZ200090, 5/7 toilets were functional, however one of them was broken (there was no water for flushing). A total of 4/7 of the schools provided adequate privacy, and there were doors that could be locked from the inside. However, two schools had cracks in the upper structure of toilet doors (KAZ200087, KAZ200090). All school toilets observed were clean, and lighting and ventilation were provided in toilets. Waste bins were also present in the school toilets.

### 3.3. Hygiene

All schools assured that water and soap were always available for handwashing.

Three quarters of parents (78.9%) answered that their child always or most of the time washed hands with soap before eating in school or after using the toilet at school. Only 2.2% of them did not know if their children washed their hands at school. However, 17.2% of parents did not answer the question.

There was a question asking the reason for not washing hands with soap at school. About half of parents (49.1%) did not answer this question, a quarter (23%) reported that they did not know, and 6% ticked the alternative “other”, where parents could indicate their own open-ended answers: Answers emerging from this included that the parents indicated that their children disinfected their hands with antiseptic. Furthermore, 8.9% of respondents answered that there was no soap or other detergents available. Some parents (6.9%) indicated that there was no or limited access to water. A mere 2.6% of the parents indicated that their child did not desire or lacked the skills to wash hands with soap. Finally, 3.4% of parents indicated that their children did not have time to wash their hands.

Parents who answered that their children washed hands most of the time indicated that reasons for not always washing was no soap (8%), no water (6.6%), no time (3.2%) and lacking desire or skill of washing hands with soap (2.6%).

Water for handwashing was provided in handwashing facilities. Of the seven schools that were observed, only one did not provide soap in the handwashing facilities. Warm water was available in four of the schools. All handwashing facilities were clean. Drying materials after handwashing were only available at two schools.

## 4. Discussion

We present the results of the study on assessing access to Water, Sanitation and Hygiene in schools of Central Kazakhstan in order to draw due attention to this issue, since the lack of official data on Kazakhstan in the WHO/UNICEF Joint Monitoring Program on WASH (JMP) is a significant problem. The research focused on studying the context with access to water, sanitation and hygiene in schools in the urban area, continuing the work of studying schools in rural areas [21,25], thereby expanding our database for the entire conjuncture in the region. By conducting this study during the COVID-19 pandemic, it was important to show not just the impact of WASH on different aspects of life, such as health and education, but to direct people to understand the significance of development of a systematic approach to water, sanitation and hygiene by clarifying the actual examples [26].

All seven schools had access to at least basic WASH services and to in-building toilets connected to the public water supply, although only 77.9% of parents answered that toilets were inside. The analysis of the data on the number and accessibility of toilets in schools showed that only 2/7 of the studied institutions fulfilled the established government norms on the ratio of number of students to number of water closets. However, it is essential to note that according to school timetables, all students are not present in classes at the same time (including during non-pandemic times), since learning operates on a two and/or three shift system. Considering these circumstances, the requirement for the number of water closets is fulfilled by 4/7 of the studied schools. Government regulation of the number of toilets in educational institutions fully justified: a study in New Zealand showed that students may refrain from visiting WC rooms due to queues and/or unwillingness to encounter acquaintances there [27]. The criterion of quantity will certainly affect accessibility. The proximity to classrooms is especially important for lower-grade children and girls because of: (1) the possibility to get to the toilet quickly if experiencing urinary urgency and (2) sense of security [28,29]. Separately, it should be noted that one of the schools (KAZ200084) additionally had a toilet not on the school premises. On the one hand, the presence of a such a spare and independent model points to the level of school improvement, but on the other hand, such locations need more serious monitoring, primarily to ensure the safety of students in every sense [30]. Nowadays, very often toilets, including those in urban areas, do not meet these requirements [29,31,32,33].

According to the results of the analysis of access to drinking water, the school administration responded that water fountains, coolers and other sources are freely available in the canteens and buffets of schools. However, this overlooks such a remarkable fact that the number of canteens in schools in this region rarely exceeds one; moreover, only 2/7 of the schools had drinking sources throughout the entire building (classrooms, corridors). During the pandemic, as part of preventive measures in educational institutions, catering rooms did not function in order to avoid an undesirable large concentration of students in one place. Current situations clearly demonstrate the advantage of systematic approach to WASH services, additionally taking into account that only half (53.2%) of the interviewed parents are sure that their children have constant and free access to drinking water. The issue of access to drinking water is not a Yes/No question, but a “How” question. Such an approach should give special priority to both unique local and global water, sanitation and hygiene problems and help to build a flexible strategy based on the human right to drinking water, considering the non-static world and economic context. The implementation of this study during the pandemic showed that the monitoring and constantly updated assessment of the access to this indicator should include the criteria of independence from various factors.

One of the criteria for access to safe water is also the issue of quality of water, because poor-quality drinking water at school can cause outbreaks of serious diseases. As mentioned earlier, schools are an important link in human socialization and require the highest quality WASH services, but when it comes to water quality research, there is much less attention paid to educational institutions [30]. To ensure safety, some (7.7%) parents prefer to provide their children with bottled water from home or stores. Furthermore, the growth of doubts about the quality of water from drinking fountains is actively manifested among schoolchildren themselves [34]. However, the school environment has a particular impact: for example, when thirsty after physical activity, students may well neglect recommendations and own beliefs and drink contaminated water [35]. Therefore, the risks of rising levels of water pollution in schools should not be solved by finding alternative sources of water. In the process of meetings with parents and administration, we made attempts to explain the importance of consolidating their joint efforts in this direction. Collaborations like these are more successful in monitoring water, sanitation and hygiene services because they follow the common goal of creating favorable conditions that promote health and mental development for students.

The WASH initiative in schools should not be terminated by provision of water facilities, but construction should mean a new beginning. The issue of sanitation, visual cleanliness, and the presence of basic amenities in WC rooms, such as the ability to close the door and retire create the necessary conditions for visiting them [36]. The cleanliness of water closets has not only a visual effect, but the systematic cleaning of surfaces in any locations significantly reduces the risks of infection and spread of various diseases [37]. The complete refusal or unwillingness of children to use toilets as needed (on time) can also lead to diseases of the urinary and gastrointestinal systems [38,39]. In this study, 60.6% of parents said that their children go to the toilet regularly, but it is worrying that 15.8% of respondents answered that their children only go when it becomes hard to endure. Moreover, 3.4% of parents responded that their children do not use school toilets; taking into account that students stay in classes up to four hours on average, this is a rather dangerous period of abstinence from a medical standpoint. The observations showed that 4/7 of the school toilets provided privacy and 7/7 of the toilets were clean; additionally, administration representatives stated that toilets were cleaned at least twice a day. Ensuring privacy and cleanliness is especially important during puberty for adolescent girls, who face many challenges to maintain feminine hygiene [40,41,42]. However, it should be borne in mind that the element of student embarrassment and long-standing taboos on discussing such sensitive topics make the assessment of access to WASH incomplete or incorrect. In this regard, explanatory work with students and surveys (including anonymous ones) provide a lot of information.

The present circumstances of the study implementation forced us to exclude our student questionnaire, i.e., the primary users of these services, which could have had a strong impact on the results. However, the chosen methodology has shown its sustainability [43]. The administration’s response stated that all water closets were fully equipped with hygiene products, but 8.9% of parents indicated that there was no soap and 6.9% of parents no water. Moreover, observations revealed that one room was not equipped with hygiene items and there was no hot water in four out of seven toilets. Our final assessment of access to hygiene indicators of WASH demonstrates (1) the consequences of poor parental involvement (23% do not know if their child uses soap in schools) in monitoring WASH services and (2) the necessity to establish or improve the previously mentioned parent–teacher collaboration. The postulate of a causal link of non-compliance with the rules of personal hygiene and an increase in the incidence of diseases is constantly brought to the attention of children [18,44,45]. At the same time, the insufficient level of access to hygiene items at school creates a paradoxical situation (discrepancy between desire and opportunity) among students, which in the future could potentially become one of the triggers for the development of unwillingness to observe hygiene rules, due to accumulated negative experiences. Returning to our research, it can be argued that the COVID-19 pandemic has refreshed unfairly sidelined questions about access to hygiene and hygiene culture in general [46,47].

The assessment of access to water, sanitation and hygiene within the framework of the SDGs showed not only the absence of the coherence and consistency of the correct implementation of WASH programs and SDG 6 in schools, but also the erroneous opinion that water, sanitation and hygiene are isolated from the educational process. While pursuing the goals of providing water and sanitation, in parallel, the necessary conditions are created for the implementation of SDG 4 to ensure inclusive and quality education [20,22]. Maintaining sanitary standards directly affects attendance, and the elimination of the externalities (lack of a toilet or unwillingness to visit it for various reasons in case of physiological need) contributes to the motivation of students to learn, freeing them from unnecessary thoughts and worries. The targets of SDG 4, SDG 6 and their clear interrelationships should first be communicated to educational staff and students through the importance of access to WASH in schools. This practice should be encouraged.

## 5. Conclusions

Central Kazakhstan is not a critical region in terms of access to water, which is proved by the study data. Nevertheless, the current assessment of access to WASH in urban schools has identified gaps (which the COVID-19 pandemic has greatly exposed) at every stage of the implementation of water, sanitation and hygiene services. Firstly, observation and analysis of questionnaires (answers and frequent ignoring of questions) revealed an insufficiently deep understanding of the importance of this issue on the part of parents and administration, which in turn naturally lays the foundation for all subsequent problems: lack of centralized control and constant monitoring, partial non-compliance with sanitary standards, and negative impact on both health and education.

Secondly, the study confirmed that the full provision of access to these services in the structure of educational institutions solves several SDG targets. However, it is not enough to know “what these tasks are” and “why these tasks are needed”, but it is essential to comprehend “how exactly these tasks need to be solved”. Timely recognition of the feasibility of a system approach to WASH not only contributes to achieving the goals of SDG 4 and SDG 6 to create an enabling environment for education and life, but also (1) makes it much easier to identify the real sources of problems by asking the right questions and (2) eliminates situations with “blind” inconsistent selection of routine tasks when decisions are made but do not bring results. Given the above gaps, in order to achieve the SDG goals in Kazakhstan by 2030, it is necessary to continue working in this direction.

## Figures and Tables

**Figure 1 ijerph-19-06438-f001:**
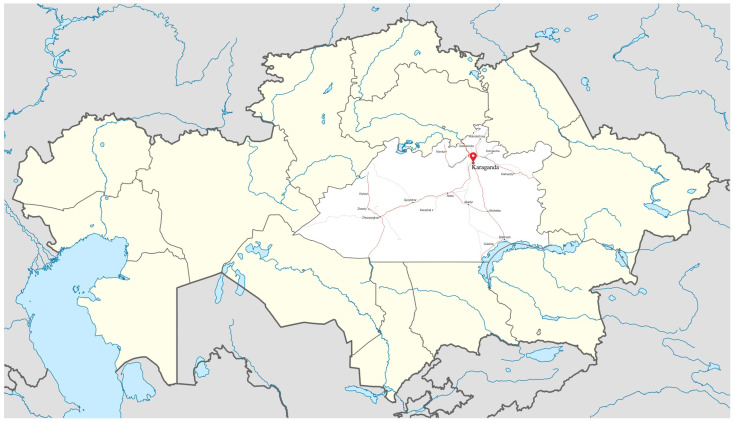
The location and borders of the Karaganda region and Karaganda city (selected area) on a map of the Republic of Kazakhstan. The figure was modified from Google maps (https://goo.gl/maps/1wGbBVcHgDUv646UA) accessed on 1 January 2022.

**Table 1 ijerph-19-06438-t001:** Selected questions from the tool WHO/UNICEF “Surveillance of water, sanitation, and hygiene in schools”.

Assessment Criteria of WASH	Questionnaire for Administrative Staff	Observation	Questionnaire for Parents
**Drinking water**			
**Availability**	Main water source	-	Availability of drinking water during the school day
	Water availability		
	Availability of drinking water supply facilities		
**Accessibility**	Specific time to drink water	-	-
**Quality of services**	Responsible person for functioning and maintenance of drinking water supply	-	-
	Measures to ensure drinking water supply		
**Education**	-	Information and educational materials about water	-
**Sanitation**			
**Availability**	Availability of the toilets	The type of the toilets	Using the school toilet
	Number of available toilets (general, for girls, for boys, for school staff)	Availability of the school toilets (not closed doors of the school toilets)	Location of the school toilet
	-	-	Reasons for not using the school toilet
**Functionality**	Problems with the functionality of the school toilets	Functional school toilet (not broken, not clogged)	-
**Privacy**	Privacy of the school toilets	Private school toilet (toilet with closing doors that lock from the inside and without large cracks in the upper structure)	-
**Accessibility**	Specific time to visit the toilet		-
**Quality of services**	Ventilation of the school toilet	Cleanness of the school toilet	-
	Enough lighting of the school toilet	The lighting of the school toilet	-
	Heating of the school toilet in the cold weather	Ventilation of the school toilet	-
	Toilet paper in the school toilet	Toilet paper in the school toilet	-
	Cleanliness of the school toilet	Waste bins in the school toilets	-
**Hygiene**			
**Availability**	Availability of water for handwashing	Availability of water for handwashing	Washing hands before eating and after toilet using
	Availability of soap for handwashing	Availability of soap for handwashing	Reason for not washing hands
	-	Availability of warm water for handwashing	-
		Availability of hand drying materials	
**Functionality**	-	Reasons for not functional handwashing facilities	-
**Quality of services**	-	Cleanness of handwashing facilities	-

**Table 2 ijerph-19-06438-t002:** Sample sizes of parents from different schools.

School	Sample Size
KAZ200084	49 (14.08%)
KAZ200085	50 (14.37%)
KAZ200086	46 (13.22%)
KAZ200087	46 (13.22%)
KAZ200088	49 (14.08%)
KAZ200089	47 (13.50%)
KAZ200090	61 (17.53%)

**Table 3 ijerph-19-06438-t003:** Availability of drinking water supply facilities.

Schools	Taps or Fountains Outside the Toilet	In the Canteen for Free	Students Bring Water from Home	Students Buy Water in the Canteen
KAZ200084		✔		✔
KAZ200085		✔		✔
KAZ200086	--	✔		--
KAZ200087		✔		✔
KAZ200088	✔		×	✔
KAZ200089	✔		✔	
KAZ200090		✔		✔

## Data Availability

Data available in a publicly accessible repository. The data provided in this study will be attached to the article.

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
