# Peer review of "Assessing Access to WASH in Urban Schools during COVID-19 in Kazakhstan: Case Study of Central Kazakhstan"

_ijerph, 2022, doi:10.3390/ijerph19116438_

Round 1

Reviewer 1 Report

It is a nice area of research, however, there are serious gaps that needs to be addressed:

  • the link of WASH in Schools and COVID in not clear. What difference it would have been if this study was conducted without COVID restrictions?
  • introduction section needs to be strengthened with additional literature review and suggest to highlight the problems and need for the study
  • in methodology, I have a strong concern, why pupils/students are not interviewed (why their parents). Although few students were attending schools, it would have made much more sense to interview them as they are the users........... how the sample size was determined?
  • discussions section needs to be completely revised as don't see it aligned with the presented results.
  • similar comment with conclusions section. It seems that some of the conclusions are not supported by results......

Author Response

Thank you for your comments and recommendations. The structure and content of the article have been greatly improved with the help of them. Shortcomings were corrected, the list of references was expanded and most of the text was rewritten. Your support has been more than significant.

Reviewer 2 Report

This is a very interesting and relevant manuscript that address an important development issue of freshwater supply and sanitation in the school environment in Kazakhstan, especially under the difficult conditions caused by the COVID-19 pandemic which challenged the sustainability of running many schools and related institutions. As the manuscript maintains, full provision of WASH facilities in schools covers several SDG challenges.  

Appropriate data have been collected and important results have been generated.  However, for the manuscript to be considered again, the authors are advised to address the following comments so that the scientific and literature context of the study are reinforced.  

comment: the current introduction is too thin. can you please place the topic in its global context before your flow of ideas proceed to your study area. there are many studies and literature points that you can bring in and this will help you to better indicate the rationale and justification of your research aim or research problem. can you please provide a literature-based argumentation on why this study is important and worthy of being investigated.  

comment: after specifying your research aim at the end of your introduction, can you please write down your research objectives so that we have an idea on what is the goal of this research? 

comment: can you please draw a proper map that shows the geographical location of your study area so that readers are afforded the opportunity to see where it is located. 

comment: in the results section, I am missing a whole section that was supposed to summarise the demographical characteristics of the respondents whether they are school children,   parents or school authorities.  It is important to provide such a section so that we can understand the characteristics of your sample of respondents. moreover, you must also explain what the demographics and socio-economic characteristics  mean in the light of your research aims or research objectives. 

comment: under the section that deal with the results, the numbering of your subsections is faulty. can you please correct the numbering sequence.. currently, you have the following numbering pattern which is wrong: 

3. Results; 3.1. Water; 3.2. Sanitation; 2.3. Hygiene

i think it must be written as follows:- 

3. Results; 3.1. Water; 3.2. Sanitation; 3.3. Hygiene

Comment:  I realise that in certain sections, especially regarding the presentation of your results, numbers are used to start new sentences. This is inappropriate. let me give you a simple example: 

page 7 of 9: lines 217-220 

'60.6% of parents assured that their children went to the toilet at school whenever they needed it. 15.8% of children went to the toilet rarely, only when it was hard to endure, 3.4% of parents replied that their children did not use it. 2.9% of parents did not know if their children used the school toilet or not. 

my suggestions are as follows:  please write the numbers in words at the beginning of the sentences and then put the numbers inside brackets. here is an example that you can follow: 

[.... About 60% (60.6%) of parents assured that their children went to the toilet at school whenever they needed it. ...  ]

[... Whereas 15.8% of children went to the toilet rarely and only when it was hard to endure, 3.4% of parents replied that their children did not use such facilities.....]

[....A very small (2.9%) percentage of parents did not know if their children used the school toilet or not...... ] 

you are also encouraged to read other published manuscripts so that you can apply this skill. 

Comment: in  certain sections of the manuscript, the english writing, grammar and punctuation is a problem in need of editorial changes. few examples are given for your attention. 

page 8 of 9; line 273: 'One of the most difficult difficulties for WASH in schools is sustainability.

Comment: The word 'difficult' and 'difficulties' cannot be allowed to follow one another like this. I suggest that you use only one of them and you reformulate the whole sentence so that it is grammatically sound.  

Comment: a major flaw is that not enough literature has been drawn in to the discussion of the findings.  it is therefore, important to bring additional literature points so that  your findings are comparable to other studies and we can clearly see their significance.  

Comment: it is not clear why the results in subsections 'section 3. 2' and '3.3' are presented without including any graphical representations or illustrations that show us the results. why is this? please include well annotated figures or tables to present your results so that we can clearly see the analysis of your results. 

the comments are given in the spirit of improving this manuscript . 

thank you for the opportunity.  

Author Response

(The authors gave the same response as above.)

Round 2

Reviewer 1 Report

The manuscript reads well.

Some of the sentences are difficult to follow............ might be good to keep it  simple. For example - Lines 164-168: "At that time, education was in a distance format, and only students in grades 1-4 could traditionally study, provided that there were a maximum of 15 pupils in a class and from 5 to 180 pupils for the entire institution, which greatly influenced the involvement of the planned number of respondents".

Using "WASH" consistently through out the manuscript instead of "Water, Sanitation and Hygiene", which has been used in several places.

Author Response

Thank you for your comments, recommendations and multi-stage support. We took everything into account and made the appropriate adjustments.
